# Empowering Pharmacists: Strategies for Addressing the Opioid Crisis through a Public Health Lens

**DOI:** 10.3390/pharmacy12030082

**Published:** 2024-05-23

**Authors:** Tamera D. Hughes, Juliet Nowak, Elizabeth Sottung, Amira Mustafa, Geetha Lingechetty

**Affiliations:** UNC Eshelman School of Pharmacy, University of North Carolina at Chapel Hill, Chapel Hill, NC 27599, USA; jnowak@unc.edu (J.N.); esottung@unc.edu (E.S.); ammustafa98@gmail.com (A.M.); geethal@ad.unc.edu (G.L.)

**Keywords:** harm reduction, community pharmacy, opioid misuse, public health, opioids, addiction, medication safety, medication-assisted treatment

## Abstract

Background: The opioid crisis in the US is a severe public health issue, prompting pharmacists to adopt various strategies for prevention, harm reduction, treatment, and recovery. Despite progress, barriers persist. Results: This commentary examines five determinants of public health in relation to pharmacist-led interventions for the opioid crisis: individual behavior, social factors, policymaking, health service accessibility, and biological/genetic considerations. Pharmacists can influence individual behavior through education and support, address social determinants like stigma, advocate for policy changes, ensure health service accessibility, and personalize opioid prescriptions based on biological factors. Conclusion: Pharmacists play a crucial role in addressing the opioid crisis by navigating these determinants. Pharmacists’ engagement is essential for reducing opioid-related harms and improving public health outcomes through advocacy, service provision, and education.

## 1. Introduction

The opioid overdose crisis in the United States is the largest drug crisis on record, with more than 500,000 lives lost in the past two decades [1,2,3]. In their latest public health efforts, pharmacists are addressing the crisis by employing several strategies that provide a broad range of prevention, harm reduction, treatment, and recovery services [4]. Even though these strategies have demonstrated efficacy [5], barriers still exist to employing the strategies.

A statement by the American Society of Health-System Pharmacists (ASHP) serves as a template for pharmacist-led public health initiatives and calls on pharmacists to develop guidelines for interventions [6]. According to the statement, five determinants of public health need to be understood to remove barriers to patient uptake. This requires examination through a public health lens, particularly according to all determinants of health in the construction of policies and the deployment of services. The five primary determinants of health outlined by ASHP that should be further researched are individual behavior, social determinations of health, policymaking, health services, and biology and genetics. This commentary aims to use ASHP’s guidelines and provide recommendations for the opioid overdose crisis by reviewing pharmacists’ role according to a range of personal, social, economic, and environmental factors that can influence opioid overdose harm reduction [7,8].

To provide context, it is important to note that the ASHP statement is just one of the significant statements made by a professional pharmacy organization in the United States addressing the opioid crisis. Other organizations, such as the American Pharmacists Association (APhA) and the National Association of Boards of Pharmacy (NABP), have also issued statements and guidelines on this issue [9,10,11]. For instance, APhA has emphasized the role of community pharmacists in opioid stewardship, advocating for increased access to naloxone and medication-assisted treatment (MAT). Similarly, NABP has focused on enhancing the use of prescription drug monitoring programs (PDMPs) and supporting legislative changes to improve opioid dispensing practices.

While the ASHP’s statement provides a detailed framework for pharmacist-led public health interventions, it distinguishes itself by emphasizing the integration of these efforts across various determinants of health. In contrast, the APhA’s approach is more focused on immediate community-based interventions, and NABP’s recommendations are geared towards regulatory and policy enhancements. This commentary will delve into these different perspectives, highlighting the unique contributions and collaborative potential of each approach in addressing the opioid crisis.

## 2. Pharmacists’ Roles in Addressing Health Determinants in the Opioid Crisis

Pharmacists play a crucial role in addressing the opioid crisis through a multifaceted approach that includes harm reduction, prevention, referral to treatment, drug regulation, policy advocacy, and fostering interprofessional relationships. Understanding and integrating the various health determinants are essential for these interventions to be effective.

### 2.1. Individual Behavior

One strategy to prevent and reduce harm in patients with OUD is to empower pharmacists to focus on their individual behavior [6,12]. ASHP emphasizes individual behavior as a crucial determinant according to which pharmacists can improve public health. Pharmacists are encouraged to discourage the use of harmful substances and acknowledge the importance of the changes substance users make in their personal lives. Accepting drug users and treating them with dignity and compassion, protecting drug users’ rights, and maintaining transparency in decision-making, whether it is successful or unsuccessful, are individual behaviors pharmacists can employ [13]. 

Pharmacists can have influence over patients with substance use disorders (SUDs). When dispensing opioids, pharmacists are responsible for naloxone education and counseling interventions [14,15]. Being that final checkpoint between a patient and their medication can help guide these individuals to a path of success. While there is no universal application of protocol or policy that outlines how to approach patients with SUDs in the United States, pharmacists should influence and counsel patients in a community setting about their medication. Pharmacists have the capability to work with local partners and resources to identify supportive services to these individuals to lessen the barriers to accessible harm reduction services for patients. Giving patients individualized care plans to protect their health while respecting their autonomy is just one way for pharmacists to make a difference in opioid overdose harm reduction.

It is empowering for patients to know their pharmacists have taken the time to validate and understand them. There is no universal application that addresses all individuals in the same way, and pharmacists need to tailor interventions and maximize the options for supportive services depending on each situation. Supporting patients who make poor health decisions by fostering individualism and accountability can lead to long-term improvements in patient outcomes. 

In closing, pharmacists can directly influence the individual behaviors that contribute to opioid misuse. To empower pharmacists in this role, specific strategies include:Education and Counseling: Provide personalized education on opioid use, safe storage, and disposal. Implement motivational interviewing techniques to support behavior change.Naloxone Training: Equip pharmacists with comprehensive naloxone training to educate patients and caregivers on its use.Screening Tools: Utilize validated screening tools to identify patients at risk of opioid misuse and provide early interventions.

### 2.2. Social Determinants of Health

Addressing social determinants involves tackling barriers that prevent optimal health outcomes. Pharmacists should engage in:

#### 2.2.1. Mitigating Barriers

The major barriers of social determinants in the context of the opioid crisis include accessibility, availability, and stigma. ASHP writes that pharmacists can screen for social determinants of health, helping to identify the major barriers that keep individuals from accessing the resources they need. Although pharmacists cannot control factors such as whether a patient is employed or the physical environment of a patient, pharmacists can play a significant role in access. Recent reports perpetuate the morbidity and mortality of opioid overdose disparities and deaths for non-Hispanic Black communities when compared to other racial/ethnic groups [1,16] and the lack of access to opioid overdose and OUD treatment, providers, and services for other marginalized populations [17,18]. Furthermore, multiple studies found that naloxone was not as accessible in rural areas, independent pharmacies, and populations where the majority are publicly insured [19,20]. And while the availability of medications is not entirely in control of the pharmacist, procuring the necessary drugs and prioritizing them in inventory can be simple ways to bridge access in neighborhoods and locations that may not necessarily have them. 

#### 2.2.2. Addressing Stigma

Stigma, on the other hand, is a more difficult barrier to address. Participants from one study described pharmacists and pharmacy technicians as the people from whom they faced the most stigma [21]. Identifying the roots of stigma and implicit bias is crucial. Therefore, pharmacists need to initiate research and curriculum analysis to restructure the way we approach addiction, especially OUD. Increasing training surrounding OUD at the student and practitioner levels may help to create a more inclusive environment. 

#### 2.2.3. Community Outreach and Education 

ASHP emphasizes how pharmacists can improve trust between individuals and the healthcare system by promoting health education, which may reduce stigma. Pharmacists have a lot of opportunity to take the lead and reduce the social obstacles associated with the opioid problem. While many patients are unaware of the resources available, pharmacists are not! Therefore, there is need for pharmacists to advocate for health services and to educate patients about the resources available to them.

To empower pharmacists in this role, specific strategies include:Community Outreach: Develop partnerships with local health organizations, community centers, and social services to enhance resource distribution and support networks for individuals struggling with opioid use.Stigma Reduction Initiatives: Organize workshops and training sessions aimed at reducing stigma within the pharmacy practice. Encourage open discussions about addiction and recovery to foster empathy and understanding.Advocacy for Policy Changes: Work with professional pharmacy organizations and health policy advocates to push for legislative changes that empower pharmacists to engage more actively in public health initiatives.Curriculum Enhancements: Collaborate with pharmacy schools to ensure that training on social determinants and stigma is a core part of the curriculum. Provide continuing education opportunities for current practitioners to stay updated on the best practices.

By addressing these aspects, pharmacists can significantly mitigate the barriers imposed by social determinants of health and contribute to more equitable opioid harm reduction efforts.

### 2.3. Policymaking

ASHP advocates for pharmacists to be involved in the development of local policies and national programs for public health legislation. Our interpretation of this statement would be utilizing pharmacists’ firsthand experience with opioid dispensing in the development and oversight of such legislation and resulting opioid policies. At the federal level, pharmacists have a *corresponding responsibility* to ensure the legitimacy of opioids and other controlled substances [22]. Unfortunately, the current language included in state opioid policies is very general and does not empower or encourage pharmacists towards proper opioid stewardship. For instance, North Carolina’s State Opioid Action Plan states three goals: expand comprehensive drug user health services to encompass multiple substances, increase healthcare services in low-barrier and non-traditional settings, and expand drug checking to prevent overdoses [23]. These general statements demonstrate a call to action, yet it is difficult to know how to act. This policy leaves many with questions such as, what multiple substances are they referring to? Who should increase healthcare services in non-traditional settings, and how? What types of non-traditional settings? What does expand drug checking really mean? This lack of clarity in the major policies prevents any entity from taking full responsibility to implement the goals. 

Widespread national and state policy support for pharmacists in overdose prevention and drug distribution has been found, particularly in dispensing naloxone [4]. Naloxone standing orders provide pharmacists with the authority to dispense naloxone without a prescription, which has potential to decrease opioid overdoses. As of July 2022, all 50 states, the District of Columbia, and Puerto Rico have some form of a naloxone access law. A total of 33 states give pharmacists the supremacy to administer naloxone under statewide standing orders, while 14 states allow pharmacists to enter into agreements with prescribers at their own volition [24]. Oregon and Idaho are the only two states presently allowing pharmacy prescription authority, including the dispending of naloxone. Despite these policy changes, making naloxone recommendations to patients who use prescription and/or illegal opioids appears to be a barrier to pharmacists [14]. Therefore, we believe that these policy changes cannot be effective without education. 

According to the FDA, pharmacists should receive training on the safe use of opioids so they can monitor patients [25]. In 2015, Virginia implemented a requirement for every pharmacist, whether dispensing opioids or not, to participate in a one-hour continuing education course on preventing opioid addiction [26]. Currently, the Virginia Board of Pharmacy has updated the requirement to 2 h [27]. Though continuing education is a requirement, implementing mandatory opioid training can help alleviate some gaps in the knowledge and care perpetuated by pharmacists. 

In facilitating policies such as standing orders for naloxone, pharmacists had to advocate. Advocacy is an incredibly crucial part of our profession and can be extended to political, justice, and geographic policies that directly influence our practice and the populations we aim to serve. Pharmacists, and even pharmacy students and residents, need to make more of an effort to engage with policymakers and institutions to better address public health crises, especially the opioid crisis. 

Pharmacists must be proactive in shaping policies that impact opioid misuse. To empower pharmacists in this role, specific strategies include:Policy Advocacy: Engage in advocacy efforts to influence legislation and regulatory changes that enhance pharmacists’ roles in opioid stewardship.Clear Guidelines: Collaborate with policymakers to ensure pharmacists’ perspectives are included in opioid-related legislation to develop and disseminate clear, actionable guidelines for pharmacists based on national and state policies.

By being actively involved in policy advocacy and development, pharmacists can play a crucial role in shaping effective opioid misuse prevention strategies and improving public health outcomes.

### 2.4. Health Services

Access to health services is a vital factor in implementing opioid crisis strategies [28]. As the most accessible healthcare professionals, pharmacists can actively participate in efforts to prevent and treat OUD and overdose, as well as providing essential medication-related services [29,30]. Pharmacists can provide counselling on opioid risks and safety, dispense naloxone, educate patients on the importance of proper medication disposal and storage, utilize prescription drug monitoring programs (PDMPs), deprescribe opioids, and provide resources for opioid misuse and addiction treatment [14].

Deprescribing, the process of tapering or stopping medications that may no longer be necessary or that may be causing harm, is an emerging health service for pharmacists in managing the opioid crisis [31]. Pharmacists, by reviewing patient medication regimens, can identify opportunities to safely reduce opioid use, particularly in patients who are at risk of adverse effects or who are using opioids in combination with other potentially harmful medications. Researchers at UNC are currently evaluating the role of pharmacists in opioid deprescribing for older adults, aiming to reduce the risk of falls and other complications [32,33,34]. This practice involves careful assessment of a patient’s clinical status, communication with prescribers, and close monitoring to ensure patient safety and the efficacy of the reduced regimen. 

Pharmacists also play a crucial role in implementing other opioid harm reduction strategies and providing the optimal care to their target population, especially in the United States. Despite their critical role, there is still reluctance among pharmacy professionals to participate in sterile syringe and naloxone dispensing [35,36,37,38]. Contextual factors influencing pharmacists’ provision and delivery of opioid harm-reducing health services should be explored to create sustainable practices that ensure adequate access to all, especially systematically minoritized and marginalized communities. For example, non-prescription syringe (NPS) sales is a service that many community pharmacies offer as a harm reduction strategy [35]. During the COVID-19 pandemic, syringe service programs (SSPs) expanded using mail-based distribution of supplies [39]. These mail-based delivery services proved hopeful for underserved individuals and individuals in low-SSP-access areas [39]. Furthermore, SSPs assist with other public health efforts and allow for accessible testing measures for HIV and Hepatitis C prevention and related services [39]. SSPs also offer the on-site distribution of medications for opioid use disorder (MOUD), which are now allowed to be prescribed through telehealth and telemedicine visits, a crucial facilitator for MOUD treatment, post 2019 [39]. Furthermore, SSPs have begun to offer mental health counseling and primary care services at their sites, a service described in the ASHP statement as a key intervention to address the opioid crisis [6].

We urge pharmacists to partner with providers, view themselves as part of the team, and communicate with patients to promote opioid harm reduction. The CDC urges pharmacists to address prescription opioid abuse and overdose [40]. Pharmacists are frequently faced with limited time and information; however, they still have a duty to assess, verify, consult, and communicate in the usual course of practice. Building and maintaining cooperative working relationships with providers can assist with alleviating some of these hurdles and improve patient outcomes while reducing opioid use disorder and overdose. After all, pharmacists are part of the team and play a crucial role in managing pain and preventing abuse. Lastly, patient education and communication are important tasks for pharmacists. Communicating with patients about the dangers of opioids should be a natural progression of our practice.

To empower pharmacists in this role, specific strategies include:Training and Education: Ensure pharmacists receive training in deprescribing techniques, including patient communication strategies and the management of withdrawal symptoms.Advocate for an Expanded Scope of Practice: Advocate for an expanded scope of practice to allow pharmacists to initiate MAT.Develop Effective Referral Protocols: Develop protocols for effective referral and follow-up processes.

By advocating for policy changes that support pharmacist-led MAT and establishing effective referral networks, pharmacists can significantly enhance the comprehensive care of patients and contribute to a reduction in opioid misuse and overdose.

### 2.5. Biology and Genetics

ASHP recommends that pharmacists learn how to apply genetic knowledge to public health issues. Although genetic testing is still not frequently utilized, opioids have documented relationships between gene phenotypes and metabolism, allowing for appropriate intervention [41,42]. If a pharmacist knows that patients are poor metabolizers or are ultra-metabolizers, either through patient or physician notification, dosage adjustments can be made, potentially decreasing the risk of overprescribing. One study looking at a personalized opioid dosing algorithm demonstrated clinically meaningful pain reduction [42].

Other interventions pharmacists can make are routine checks for interactions with opioids. Although pharmacists already utilize their clinical skills in checking drug interactions, we call for pharmacists to take action to critically think about patient-specific factors for anyone who has a prescription for opioids. Furthermore, ongoing education and access to advanced clinical tools are necessary to optimize this practice, particularly in the context of polypharmacy involving opioids. For example, the concurrent use of opioids with other sedating medications such as benzodiazepines pose a significant risk in the geriatric population [43,44]. As a result, pharmacists should continue to consider age and other biological components when validating and dispensing these medications. Researchers at UNC are currently evaluating the role of pharmacists in opioid deprescribing for older adults, with the overall goal of reducing harm due to falls [32,34].

Educating people on the role of genetics and metabolism may urge patients to advocate for more personalized medicine. As the field of pharmacogenetics continues to grow, pharmacists can expand their scope of practice to acknowledge these influencing factors on opioid prescribing. Further research will need to explore sustainable payment models surrounding genetic testing and its accessibility to all populations. By leveraging pharmacogenomics and staying abreast of the latest research findings, pharmacists can optimize opioid prescribing and management to meet individual patient needs. To empower pharmacists in this role, specific strategies include:Incorporate Pharmacogenomics Training: Integrate pharmacogenomics education into PharmD programs and continuing education curricula to equip pharmacists with the necessary skills and knowledge to apply genetic testing in practice effectively.Promote Research Participation: Encourage pharmacists to engage in research endeavors focusing on personalized medicine approaches, including pharmacogenomics, to contribute to the advancement of evidence-based practice.

By embracing personalized medicine approaches grounded in pharmacogenomics, pharmacists can optimize opioid therapy, mitigate potential risks, and improve patient outcomes while advancing the field of pharmacy practice.

In conclusion, pharmacists play a pivotal role in addressing the multifaceted challenges presented by the opioid crisis, leveraging their unique position as accessible healthcare professionals to enact meaningful change. By adopting a comprehensive approach that encompasses individual behavior, social determinants of health, policymaking, health service accessibility, and biological/genetic considerations, pharmacists can effectively contribute to harm reduction, prevention, treatment, and recovery efforts. Empowering pharmacists through education, advocacy, and collaboration is essential to optimize their impact in combating the opioid crisis and improving public health outcomes. Figure 1 highlights a summary of the public health strategies to promote pharmacist engagement in the opioid crisis, providing a roadmap for collective action and progress in addressing this critical public health issue.

## 3. Conclusions

Pharmacists are being increasingly encouraged and empowered to lead public health efforts to address the opioid crisis. This commentary explores the five determinants of health and identifies strategies for pharmacists to employ in opioid overdose harm reduction. Pharmacists can improve the negative consequences of the opioid crisis by acknowledging and supporting individual behaviors; acting against adverse social determinants of health; engaging in advocacy and policy; developing and implementing harm reduction services; and engaging in public-health-related research and education programs. These strategies should be explored by pharmacists to impact public health and reduce opioid-related harm in the US.

## Figures and Tables

**Figure 1 pharmacy-12-00082-f001:**
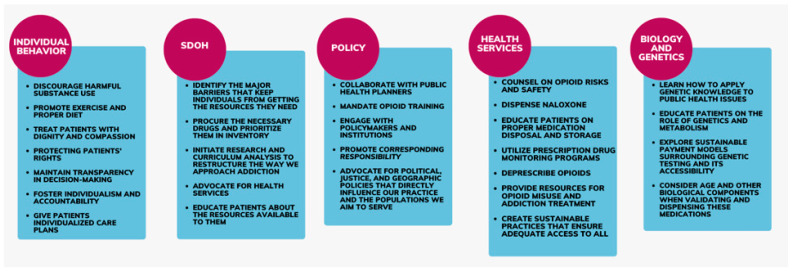
Public health strategies to promote pharmacist engagement in opioid overdose harm reduction.

## Data Availability

The data supporting this study are included within the article.

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
