# Peer review of "Empowering Pharmacists: Strategies for Addressing the Opioid Crisis through a Public Health Lens"

_pharmacy, 2024, doi:10.3390/pharmacy12030082_

Round 1

Reviewer 1 Report

Comments and Suggestions for Authors

Overall this manuscript is an important addition tot the statement released by the American Society of Health-System Pharmacists. I have a few questions and concerns outlined below. Many are around the theme tone and direction with respect to the commentary:

 - Not sure why there is a "methods section" in the abstract (I guess it's part of the journal format) but it seems out of place when there isn't a methods section in the body of the article

 - The commentary is about the American Society of Health-System Pharmacists - is this the first statement by such a body in the US? Have other organizations made similar, or different statements? This context would be useful in the introduction and if other such statements exist, a brief compare and contrast would helpful as well

 - the title and the title for section 2 are confusing. Section 2's title states that the section is about health determinants in pharmacist-led harm reduction, but the manuscript title is about empowering pharmacists, via public health, about the opioid crisis, which involves harm reduction but also prevention referral to treatment, etc. but also drug regulation, policy, and interprofessional relationships. 

 - subsections in section 2 vary with respect to tone and direction. For example, I found some of the content under "individual behaviour" good but fairly generic and not specific enough to "empower" pharmacists. In contrast, the policymaking section specifically criticizes an example of "general statements" in the NC opioid action plan and how that can lead to confusion around specific actions pharmacists can take. 

 - there is a mix of calls for more education, advocacy, and increased community engagement, and better clinical practice but these are not present in all sections, this results in a lack of cohesion from section to section

 - There is a "call" for checking opioid drug interactions but that seems like a minimum competency that every pharmacist should already be doing?

 - the content is great and this certainly a useful commentary on the statement by the American Society of Health-System Pharmacists, however I feel that the overall tone is too passive and most sections could be improved by more specific action items (whether that's changes to PharmD curricula, changes in scope of practice, regulation, etc.) (I understand that this quite a subjective critique!)

Author Response

  1. Abstract Methods Section: We have revise the abstract.
  2. Context of ASHP Statement: We have added more context in the introduction about the ASHP statement, its significance, and if similar statements have been made by other organizations.
  3. Title and Section Titles Clarity: We have revised the title and section titles to ensure they accurately reflect the content and provide clarity on the focus of each section.
  4. Consistency in Tone and Direction: We have made minor changes, ensuring to maintain consistency in tone and direction throughout the manuscript. We have also added recommendations to each section.
  5. Minimum Competency vs. Call for Action: Actionable recommendations for improvement have been included.

Reviewer 2 Report

Comments and Suggestions for Authors

I read with interest the opinion paper titled "Empowering Pharmacists: Strategies for Addressing the Opioid Crisis Through Public Health Lens"

Overall comment: The results must be toned down. Those is strong opinative and you provide no strong evidence that pharmacists could do everything you list for. 

To provide those results "Pharmacists can influence individual behavior through education and support, address social determinants like stigma, advocate for policy changes, ensure health service accessibility, and personalize opioid prescribing based on biological factors.", you should provide references, examples, or previous actions that show that this could be real. Otherwise this is only an intention. 

Reference 5, that you state to share best pharmacist practices, have no information (that I found) regarding pharmacists. Please clarify the refecence used. 

Line 27-28 - Author state "Many of these strategies (applied by pharmacists) resulted in lower overdose mortality, significant reductions in the number of emergency department visits, and a decline in the receipt of opioid prescriptions. - I checked the references and none of those talk about pharmacists. How do you conclude that the stategies applied for pharmacists could have any improvement?

Whats the basis of Image 1? Provide references. 

Paragraphs between line 55 and 68 should be justified with several references. None was found. 

Conclusions should be toned down. Eg: "Pharmacists are leading public health efforts to address the opioid crisis." Are them leading? Whats the evidence? This is very opinative. They could have a good role, that I didnt found that those professionals already have. But conclude that are leading is strongly biased. This is, for sure, a topic that should be discussed in multidisciplinary teams. As examples, the prescription efforts will be for sure from doctors, and the counselling of opioids use will be for sure shared between pharmacists and pharmacy technicians. 

Please discuss the role of desprescribing opioids - this is indeed something that pharmacies, with all healthcare professionals there, as pharmacists and pharmacy technicians, could lead in the future.

Please discuss the role on monitoring opioids use - this is indeed something that pharmacies, with all healthcare professionals there, as pharmacists and pharmacy technicians, could lead in the future.

What the reccomendations for pharmacists to the counseling of opioids risk?

What the strategies to be take on action to reduce misuse, abuse or medication errors on opioids?

As a minor comment from the formatting, references [1], [2], [3], should be merged into [1-3]

Author Response

  1. Provide References and Evidence: We have updated references and evidence.
  2. Clarify References: References have been updated. Thank you for your attention to detail.
  3. Tone Down Conclusions: We have updated this to show that pharmacists are being encouraged and empowered to lead these efforts. We believe this is also reflected throughout the text.
  4. Discuss Deprescribing and Monitoring Opioids: Deprescribing was initially discussed in the manuscript. However, we do believe that we should add additional discussions on the role of pharmacists in deprescribing opioids and monitoring opioid use, as suggested. It is now also in the health services section.

Reviewer 3 Report

Comments and Suggestions for Authors

In the appended file.

Author Response

  1. Include Brief Description of US Pharmacists' Role: We have added a brief description of the tasks, rights, functions, and legal aspects of US pharmacists in the distribution of opioids to provide context for readers outside the US.
  2. Clarify Sections and Points: We have updated the sections and hope they flow better.
  3. US-specific Examples: We have updated the manuscript to reflect the focus is on US pharmacies and pharmacists.
  4. Pharmacists' Role in Leading Public Health Efforts: We have updated this to show that pharmacists are being encouraged and empowered to lead these efforts. We believe this is also reflected throughout the text.

Round 2

Reviewer 2 Report

Comments and Suggestions for Authors

The authors adressed the requested changes. No further comments.